

# Eliciting and modeling emotional requirements: a systematic mapping review

Mashail N. Alkhomsan[1,2], Malak Baslyman[1,3] and Mohammad Alshayeb[1,4]

[1] Information and Computer Science Department, King Fahad University of Petroleum and Minerals, Dhahran, Saudi Arabia

[2] Computer and Information Sciences Department, Jouf University, Sakaka, Saudi Arabia

[3] Interdisciplinary Research Center for Finance and Digital Economy, King Fahad University of Petroleum and Minerals, Dhahran, Saudi Arabia

[4] Interdisciplinary Research Center for Intelligent Secure Systems, King Fahad University of Petroleum and Minerals, Dhahran, Saudi Arabia

## ABSTRACT

**Context**. Considering users' emotions plays an extremely crucial role in the adoption and acceptance of recent technology by the end user. User emotions can also help to identify unknown requirements, saving resources that would otherwise be wasted if discovered later. However, eliciting and modeling users' emotional requirements in software engineering is still an open research area.

**Objective**. This systematic mapping review analyzes emotional requirements (ER) practices in software engineering from two perspectives: elicitation and modeling. For elicitation techniques, we investigate the techniques, evaluation methods, limitations, and application domains. For modeling techniques, we examine the modeling languages, analyses, limitations, and domains.

**Method**. We systematically reviewed studies on emotional requirements engineering published between 1993–2023 and identified 46 relevant primary studies.

**Results**. A total of 34 studies investigated ER elicitation techniques, five examined modeling techniques, and seven covered both. Illustrative case studies were the main evaluation method for proposed elicitation techniques. Identified limitations include time consumption and extensive human involvement. The dominant application domains were healthcare and well-being, and game development.

**Conclusion**. This review summarizes the current landscape of emotional requirements research, highlighting key elicitation and modeling techniques, evaluations, limitations, and domains. Further research can build on these findings to advance emotional requirements practices in software engineering. Future research may address (1) managing conflicting emotional requirements across users, (2) evaluating the value and impact of considering emotional requirements during the development and (3) Modeling and analyzing emotional requirements in relation to other requirements.

Corresponding author
Mashail N. Alkhomsan,
g201901710@kfupm.edu.sa

# INTRODUCTION

The increasingly competitive software industry requires rapid adaptation to evolving market needs. To develop products that customers love and adopt, providers must deliver exceptional user experiences (UX). This requires engineers to prioritize UX early in development through effective requirements engineering (RE). RE encompasses determining user needs and defining product functionality and quality. For superior UX, RE should go beyond basic requirements to capture human factors like emotions and values that attract and satisfy users. Emotions are the main dimension that defines UX, and it plays a significant role in the acceptance and adoption of innovative technology by end users (*Law et al., 2008*; *Levy, 2020*). Considering them reveals latent needs (*Lopez-Lorca et al., 2014*) and prevents wasted resources after release. By incorporating emotions and values, RE can shape products into rewarding long-term user experiences. Though complex, the sensitive and systematic understanding of human dynamics builds connections between users and technology. This deeper knowledge encourages innovation and competitive advantage.

Recent research papers have examined the motivations for integrating emotions into requirements engineering. By determining the emotional experiences users wish to elicit or avoid, designers can anticipate human emotional reactions and mitigate unintended negative effects through specific design choices. This approach benefits persuasive technologies aimed at influencing behavior change, such as e-health interventions seeking to motivate healthier lifestyle habits (*Fogg, 2002*). The widespread adoption of social software provides further rationale for incorporating human emotion into requirements engineering (RE) processes. Designing social applications that foster active engagement and feelings of belonging necessitates understanding social emotions, such as empathy, to build meaningful connections between users (*Sutcliffe, 2011*; *Mooses et al. , 2022*). However, the elicitation of users' emotional requirements remains an underexplored area demanding greater focus within software engineering research and practice.

This research seeks to identify, assess, and synthesize existing literature on emotional RE elicitation and modeling following guidelines established by *Kitchenham & Brereton (2013)*. This research study aims to address the following research questions:

- What elicitation techniques have been proposed for eliciting emotional requirements?
- What are the methods and approaches used to validate the techniques' effectiveness in collecting emotional requirements?
- What are the benefits and limitations of emotional requirements eliciting/modeling techniques identified in the literature?
- What are the emotional requirements modeling languages and representation formats that have been investigated in the studies?
- What are the application domains for which emotional requirements have been elicited and represented?

Emotional requirements engineering is an emerging area within human-centered design and software development. Although emotions powerfully influence how people perceive,

choose, and interact with technology, requirements engineering methodologies typically focus on more rational and functional needs while overlooking emotional ones. While research has examined emotions in requirements engineering broadly, there remains a need to specifically synthesize findings on emotional requirements elicitation and modeling techniques. This represents an important gap, as a focused understanding of existing emotional requirements elicitation and modeling approaches is crucial to mapping the landscape and enabling continued progress. Furthermore, this review aspires to understand how emotions are defined and operationalized, which is particularly crucial for examining an interdisciplinary topic like emotions in requirements engineering, where conceptual clarity is needed to integrate perspectives from affective science, psychology, and software engineering. This review provides a targeted investigation of the current state of research and open questions about emotional requirements elicitation and modeling. The results will benefit researchers and practitioners in human–computer interaction, user experience design, software engineering, and related fields. Practitioners may improve their understanding of emotional requirements and their ability to elicit and model them. Researchers can gain a comprehensive view of techniques and outstanding needs to guide future emotional requirements research directions.

# RELATED WORK

Emotional requirements engineering is an emerging area gaining increasing focus. The role of users' emotions is being explored for several reasons within software engineering research and practice. While numerous systematic reviews have examined requirements elicitation and modeling methods broadly, few have centered specifically on the emotions dimension. This systematic mapping review examines applications, challenges, techniques, and outcomes surrounding emotional requirements elicitation and modeling. The following subsections highlight several systematic reviews related to the process of requirements elicitation and requirements modeling.

## Requirements elicitation techniques

The requirements elicitation process is the first and most significant step in the requirements engineering process. Having the elicitation conducted incorrectly will lead to low-quality products, delayed deliveries, or high expenses. *Wong, Mauricio & Rodriguez (2017)* applied the systematic literature review (SLR) to gather and evaluate the available aspects that have been covered by the different requirements elicitation approaches. These aspects are classified to cover type of contribution, level of automation, knowledge reuse, importance of human factors, collaborative approach, and types of projects. The authors also identified the activities of the requirements elicitation process as covered by the different approaches, such as identify requirements, document, and refine, and identifying factors influencing the requirements elicitation process positively or negatively, such as different stakeholders' perspectives and the project complexity.

*Pacheco, García & Reyes (2018)* conducted a systematic review of the set of validated elicitation techniques used between 1993 and 2015 to investigate effective elicitation techniques used at that time. It describes which elicitation techniques are effective and

in what circumstances they are most effective. Among other factors, the results were determined by the type of product to be developed, stakeholder characteristics, and the type of collected information.

*Iqbal et al. (2023a)* conducted a systematic mapping focusing specifically on categorizing and analyzing the state-of-the-art research on emotions in RE, providing an overview of the field. Also, it elucidates how emotions are elicited and represented across different RE phases. However, the scope was broad, and additional reviews building a focused understanding of emotional requirements elicitation or modeling techniques alone could strengthen practice.

## Requirements modeling techniques

*Yang et al. (2014)* systematically investigated the research literature on requirements modeling and analysis for self-adaptive systems. The authors summarized the state-of-the-art research trends and categorized the modeling methods used and relevant RE activities. A total of 16 modeling methods were identified to be used in 11 RE activities. They also categorized the quality attributes related to self-adaptive systems and application domains that can assist researchers and practitioners in choosing appropriate demonstrations and designing reasonable experiments.

These studies examined elicitation techniques and modeling languages with functional and non-functional requirements, as seen in Table 1. However, unlike prior reviews that either focus on general requirements elicitation techniques or provide a broad overview of emotions in requirements engineering, our systematic mapping study offers an in-depth, focused synthesis specifically targeting practices and research on eliciting and modeling emotional requirements. Therefore, we were motivated to conduct this study as emotions have a different and complex nature, and elicitation techniques and modeling would be different from those used for other types of requirements.

## METHODOLOGY

This section describes the methodology for conducting the systematic mapping review. It reports the research questions, search strategy, quality assessment approach, data extraction process, and synthesis techniques to be utilized.

### Research questions

Based on study objectives, five research questions (RQs) grouped under two categories were developed: (1) Requirements elicitation techniques and (2) emotional requirements modeling. Table 2 presents the research questions and their motivation.

### Search strategy

The search strategy phase involves identifying search terms, selecting resources, and explaining the search process for retrieving primary studies.

#### *Search string*

Search terms were formed by decomposing the research questions into main keywords: emotion, requirements, and engineering. Alternative spellings and synonyms were examined for each keyword. The resulting search string is presented in Table 3.

**Table 1 Comparative analysis of related work.**

| Study | # of studies | Contribution | Focus on emotional requirements | Focus on eliciting | Focus on modeling |
|---|---|---|---|---|---|
| *Wong, Mauricio & Rodriguez (2017)* | 42 | Synthesized key aspects and activities in general requirements elicitation | No | Yes | No |
| *Pacheco, García & Reyes (2018)* | 140 | Identified the most effective general elicitation techniques | No | Yes | No |
| *Iqbal et al. (2023a)* | 58 | Provided a broad literature review of emotions in requirements engineering | Yes | Partially (among other topics) | Partially (among other topics) |
| *Yang et al. (2014)* | 101 | Reviewed requirements modeling methods for self-adaptive systems | No | No | Yes |
| Our review, 2023 | 42 | In-depth focused synthesis of techniques for eliciting and modeling emotional requirements | Yes | Yes | Yes |

**Table 2 Systematic mapping questions.**

| No | Question | Motivation |
|---|---|---|
| RQ1 | What elicitation techniques have been proposed for eliciting emotional requirements? | This question aims to identify the elicitation techniques that have been used to elicit emotional requirements. |
| RQ2 | What are the methods and approaches used to validate the techniques' effectiveness in collecting emotional requirements? | This question aims to investigate the approaches and methods used by the researchers in validating elicitation techniques. |
| RQ3 | What are the benefits and limitations of emotional requirements eliciting and modeling techniques that have been identified in the literature? | This question aims to investigate the strengths and weaknesses that promote/hinder the use of emotional requirements. |
| RQ4 | What are the emotional requirements modeling languages and representation formats that have been investigated in the studies? | This question aims to extract the emotional requirements modeling languages and identify their overlaps and contributions to modeling emotional requirements. |
| RQ5 | What are the application domains for which emotional requirements have been elicited and represented? | This question aims to explore the applications where emotional requirements elicitation techniques are applied the most. This will shed light on several issues, such as what are the domains that tend to consider emotional requirements and what elicitation techniques may be reused or improved for similar domains. |

*Selected databases*

Four electronic search engines from the most relevant software engineering sources were selected to search for the papers: ACM Digital Library, IEEE-Xplore, Scopus, and ISI Web of Knowledge. We selected those four databases because they are recognized as the most relevant resources in software engineering, including requirements engineering, user experience, and human factors. Table 4 depicts the databases and retrieval studies after using the search string in the four electronic databases.

*Search process*

The search process was executed in three phases, as depicted in Fig. 1:

- **Search phase.** Search the selected electronic databases to find relevant studies using the search string based on the abstract and title.
- **Screening phase.** During the screening phase, 239 non-duplicated studies were assessed for eligibility. To identify duplicate studies, automated detection in Excel was used with manual review. The retrieved studies underwent two rounds of screening. In the first round, titles and abstracts were examined to determine whether studies fulfilled the inclusion and exclusion criteria, yielding a relevant subset for the review. The second round entailed a full-text assessment of each study to ascertain eligibility as per the inclusion and exclusion criteria, thus generating the final subset of relevant studies for the review.
- **Evaluation phase.** The included studies will be evaluated using quality assessment criteria.
- **Final phase.** The studies that pass the quality assessment will be the primary studies. In addition, references to the primary studies will be tracked to find more relevant ones. The tracked studies will be examined against inclusion and exclusion criteria and then assessed using quality criteria.

This multi-stage approach aimed for a broad initial search followed by progressive filtering to obtain primary studies most directly and thoroughly addressing the topics of interest. The search and screening phases cast a wide net to capture all potentially useful literature. The evaluation and final phases then refined results by systematically assessing relevance and quality, retaining only the most significant contributions meeting standards for empirical research and depth of insight. This process seeks to balance an exhaustive search with a filtered result set focused on the available evidence at the highest level of rigor and relevance.

## Selection criteria

By applying well-defined selection criteria, the most relevant and rigorous literature can be identified from an initial broad search. The inclusion and exclusion criteria serve as a mechanism for systematically filtering a large result set down to a focused subset of primary sources. Papers satisfying all inclusion criteria were included, and papers meeting any exclusion criteria, presented in Table 5, were excluded. Database searches were restricted to papers published in the date range of 1993–2023. This excluded any papers published outside this date range. The primary focus is on software engineering

**Table 3** Search string.

| Keyword | Strings |
|---------|---------|
| Requirements | (requirement* OR need* OR demand* OR request* OR necess*) AND |
| Emotional | (Emotion* OR sentiment OR affect OR feel* OR Mood) AND |
| Engineering | (Engineering OR elicit* OR collect* OR gather* OR model*) |

**Table 4** Database retrieval results.

| Database | IEEE | ACM | Scopus | Web of Science |
|----------|------|-----|--------|----------------|
| # Retrieved Studies | 106 | 6 | 102 | 100 |

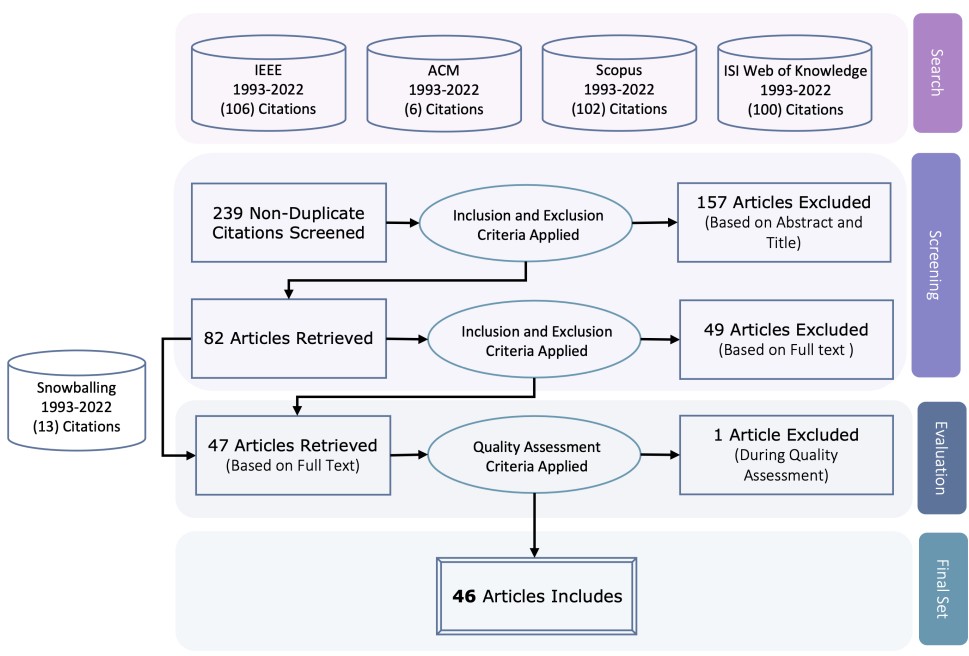

**Figure 1** Search process.

with an emphasis on emotional requirements. Also, during the full-text review, papers were assessed to confirm they addressed eliciting or modeling emotional requirements. For example, a paper may have proposed a new requirements elicitation technique and briefly noted it could be useful for emotional requirements but did not provide in-depth discussion.

## Quality assessment

The quality of the included studies was evaluated based on five criteria, shown in Table 6. Each criterion was scored on a scale of 0 to 1, with studies fully meeting the criterion

**Table 5  Selection criteria.**

| No. | Inclusion criteria | No. | Exclusion criteria |
|---|---|---|---|
| IC1. | The paper should be published between 1993 and 2022. This is because the first RE symposium was held in 1993. | EC1. | Dissertations/theses, slide presentations, personal opinions, points of view, or conference reviews. |
| IC2. | The primary focus should be on software engineering, and the study should emphasize eliciting or modeling emotional requirements. | EC2. | Systematic literature reviews and literature surveys. |
| IC3. | The paper should answer one or more of the systematic mapping questions. (Discuss eliciting or modeling emotional requirements). | EC3. | The paper is not written in English. |

**Table 6  Quality assessment criteria.**

| No. | Quality assessment |
|---|---|
| QA1 | The paper has clearly defined objective(s) |
| QA2 | The paper has reported the research methodology in a clear and coherent manner. |
| QA3 | The paper describes the approach used to propose or customize elicitation/modeling techniques in detail. |
| QA4 | The paper validated or evaluated the proposed or the customized elicitation/modeling techniques. |
| QA5 | The paper stated the results clearly. |

receiving a score of 1, studies partially meeting the criterion receiving a 0.5, and studies not meeting the criterion receiving a 0. The criteria focus on the quality and rigor of the study design, methodology, validity of results, and adequacy of reporting. A threshold score of $\geq$ 2.5 was used to identify high-quality primary studies for inclusion. This threshold ensures only studies that sufficiently meet the majority of quality criteria are used for data analysis and synthesis.

## Data extraction

A data extraction form was tailored to fulfill the identified RQs and will be filled out for each included study. This form ensures that all relevant information will be obtained from each study, which makes it possible to analyze and compare the results. Data will be extracted into three categories: research areas and directions, requirements elicitation techniques, and emotional requirements modeling. As a first step, we will collect metadata for each paper, including the title, authors, publication year, type of publication, and venue for statistical investigation. Additionally, we will extract the main gap of each paper and the future directions (if any), so we can analyze the gaps and future directions. Afterward, we will extract the data to answer questions related to the emotion requirements elicitation techniques, including the approach to proposing the technique, the validation method, the limitations and benefits, and the application domains. Finally, we will identify the emotional requirements modeling languages, contributions, limitations, and application

**Table 7  Data extraction sheet contents.**

Data extraction log
- Data extractor
- Data reviewer
- Paper information
- Study ID
- Authors
- Title
- Abstract
- Publication year
- Publication venue
- Publisher
- Publication type

Emotional Requirements Elicitation
- Elicitation Technique
- The approach in proposing or customizing the technique
- Validation Method
- Benefits and Limitations
- Application Domains

Emotional Requirements Modeling
- Emotional Modeling Language
- Analysis (if any)
- Benefits and Limitations
- Application Domains

domains if they are applied in a specific context. The extracted data will be organized using an Excel sheet, as seen in Table 7, for further data synthesis.

## Data synthesis strategy

Extracted data from the retrieved studies will be synthesized to extract outcomes using the narrative synthesis method (*Popay et al., 2006*). The data will be organized into three main themes. The first theme is to address emotional requirements elicitation techniques, validation methods, benefits and limitations, and application domains. The second theme is to investigate the emotional requirements modeling languages, their overlaps and contributions, and limitations and application domains. The third theme includes research areas, research gaps, and future directions from the literature. Finally, we will determine how best to organize and visualize the data (*i.e.,* figures/tables) based on the findings at the review stage.

## Study validation

A pilot run of the data extraction process and templates was conducted on five papers by one author to validate the study methods. This resulted in modifications to the templates and the development of guidelines for defining keywords likely to arise in selected papers. These steps improved consistency in coding and data capture. Additionally, two authors reviewed the retrieved studies, primary source selection, and data extraction. This validation helped

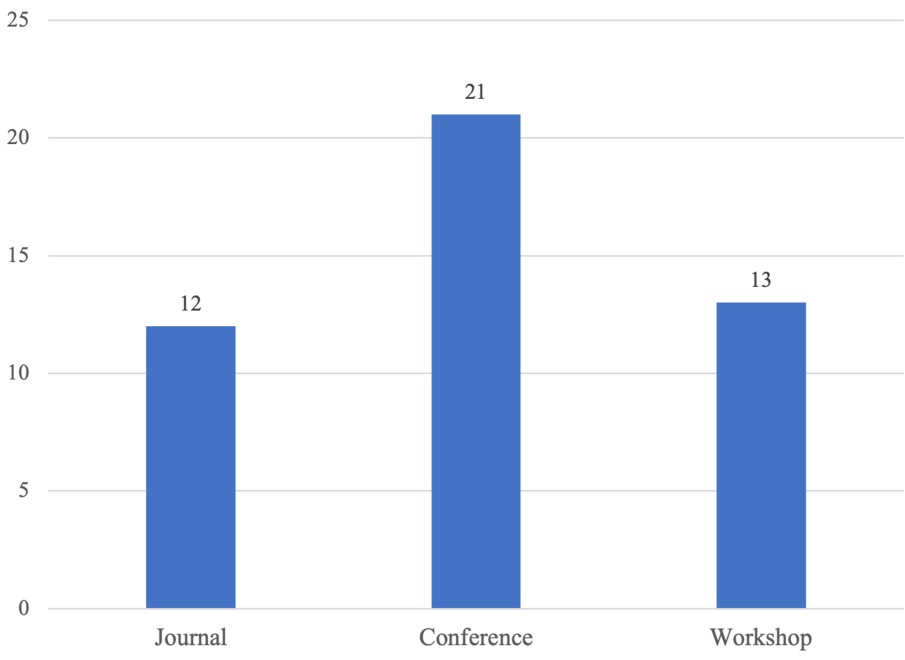

**Figure 2** **Distribution of papers by venue types.**

resolve any disagreements, ensuring consensus and reliability in the systematic search and review process.

## RESULTS

### Overview

The study includes 46 primary studies, as indicated in Fig. 1, which represents the number of studies examined during various phases of the mapping study. The list of the 46 primary studies is represented in the Appendix. The distribution of these papers across the venue types indicates that the emotional requirements area has grown considerably. As seen in Fig. 2, most primary studies (21 out of 46) are published in conference proceedings. As for the publications over the years, Fig. 3 gives an overview of the number of papers published in Emotional Requirement Engineering over the years. The number of publications in Emotional requirements is at its peak in 2019–2023. Analyzing recent published studies' motives can help in explaining the reasons why emotional requirements have gained significant attention lately. Most of the studies published between 2019–2023 aimed to increase user acceptance and trust towards new IT solutions, such as smart home systems for the elderly (*Curumsing et al., 2019*).

### Emotions definitions in requirements engineering

Understanding how emotions are defined and operationalized is particularly crucial for examining an interdisciplinary topic like emotions in requirements engineering, where conceptual clarity is needed to integrate perspectives from affective science, psychology, and software engineering. To analyze how emotions have been characterized and incorporated

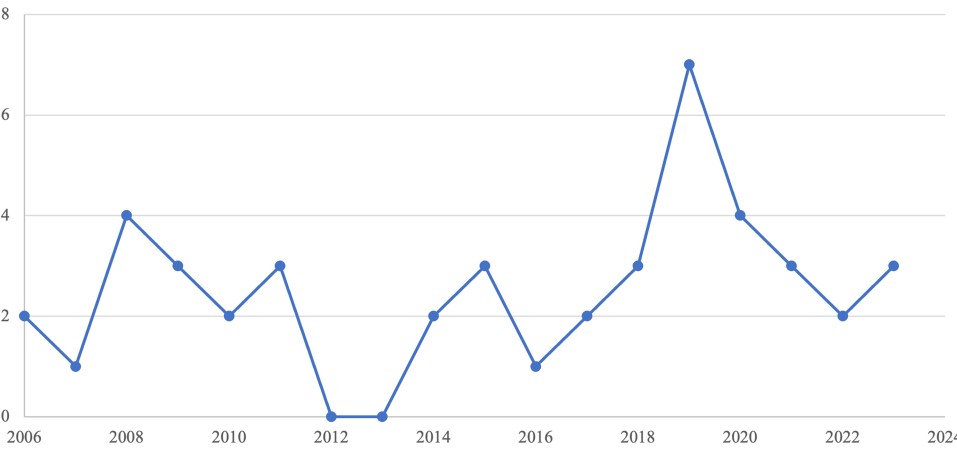

**Figure 3** Publication trends over the year.

**Table 8** Emotions definitions groups.

| | |
|---|---|
| Emotional valence dimension | *Sutcliffe (2011)*; *Izard (1991)*; *Jiang & Li (2020)*; *Thew & Sutcliffe (2018)*; *Abdullah et al. (2020)*; *Yeewai et al. (0000)*; *Stade et al. (2019)*; *Jackson & Norta (2020)*; *Zhou, Jianxin Jiao & Linsey (2015)*; *Mulvenna et al. (2017)*; *Levy (2020)*; *Maier & Berry (2017)*; *Sutcliffe (2012)*; *Cheng et al. (2023)*; *Bolchini, Garzotto & Paolini (2007)*; *Thew & Sutcliffe (2008)*; *Curumsing et al. (2015)*; *Yu et al. (2021)* |
| Valence-arousal model | *Sutcliffe (2011)*; *Russell (1989)*; *Alkhomsan, Baslyman & Alshayeb (2022)* |
| Appraisal theory | *Roseman & Smith (2001)*; *Sherkat et al. (2018)*; *Callele, Neufeld & Schneider (2009a)*; *Callele, Neufeld & Schneider (2008c)*; *Callele, Neufeld & Schneider (2006)*; *Callele, Neufeld & Schneider (2008a)*; *Callele, Neufeld & Schneider (2009b)*; *Callele, Neufeld & Schneider (2008b)* |
| Theory of constructed emotion | *Taveter et al. (2019)*; *Taveter & Iqbal (2021)*; *Iqbal et al. (2023b)* |
| Norman's three levels of design | *Miller et al. (2015)* |

across the selected studies, we have summarized the definitions of emotion mentioned in the 46 studies. Table 8 categorizes definitions and lists supporting studies. Although some definitions were adopted more frequently than others, these frequencies should not be considered indicative, as many of the studies were conducted by overlapping research groups.

- **Group 1: Emotional valence dimension**

The valence dimension categorizes emotions as positive or negative (*Izard, 1991*). Studies (*Sutcliffe, 2011*; *Jiang & Li, 2020*; *Thew & Sutcliffe, 2018*; *Abdullah et al. , 2020*; *Yeewai et al., 0000*; *Stade et al., 2019*; *Levy, 2020*; *Zhou, Jianxin Jiao & Linsey, 2015*; *Mulvenna et al., 2017*; *Levy, 2020*; *Maier & Berry, 2017*; *Sutcliffe, 2012*; *Cheng et al., 2023*; *Bolchini, Garzotto & Paolini, 2007*; *Thew & Sutcliffe, 2008*; *Curumsing et al., 2015*; *Yu et al., 2021*) underscore considering emotion valence in requirements engineering, product design, and user

experience. Analyzing valence provides insight into the potential impact of emotions on users.

- **Group 2: Valence-arousal model**

The valence-arousal model classifies emotions based on their positive/negative valence and high/low arousal (*Sutcliffe, 2011*; *Russell, 1989*). Study (*Alkhomsan, Baslyman & Alshayeb, 2022*) adopts this approach, discussing how software interactions can evoke emotions with different valence-arousal combinations. While universally defining emotional requirements is challenging, (*Alkhomsan, Baslyman & Alshayeb, 2022*) attempts to formally define "emotional requirements" using class diagrams. The proposed metamodel captures the key components of emotional requirements to represent user needs meaningfully. It consists of three main classes: User, Emotion, and Condition. The metamodel associates emotional requirements with other requirement types like functional and non-functional. This allows extending any requirement to include emotional aspects.

- **Group 3: Appraisal theory**

Appraisal theory is a psychological theory (*Roseman & Smith, 2001*) proposing that emotional responses are shaped not just by situations themselves but by how individuals interpret and understand those situations. The study that explicitly adopted this theory is (*Sherkat et al., 2018*), defining emotions as intuitive reactions towards a product or service based on appraising its ability to meet personal concerns or goals. This cognitive interpretation can lead to positive or negative emotional responses. While not mentioned explicitly, studies (*Callele, Neufeld & Schneider, 2009a*; *Callele, Neufeld & Schneider, 2008c*; *Callele, Neufeld & Schneider, 2006*; *Callele, Neufeld & Schneider, 2008a*; *Callele, Neufeld & Schneider, 2009b*; *Callele, Neufeld & Schneider, 2008b*) align with this perspective in their conceptualization of emotions as context-dependent outcomes of subjective evaluative processes.

- **Group 4: Theory of constructed emotion**

The theory of constructed emotion states emotions arises from processing goals, experiences, and situational factors. Studies (*Taveter et al., 2019*; *Taveter & Iqbal, 2021*; *Iqbal et al., 2023b*) refer to this theory.

- **Group 5: Norman's three levels of design**

Study (*Miller et al., 2015*) relates emotions to Norman's reflective design level. While Visceral and Behavioral Levels: Focus on software functionality, usability, and performance. Reflective Level: Introduces emotions as "properties desired at a reflective level" regarding cultural meaning and personal resonance.

## Emotional requirements elicitation techniques

This section addresses the elicitation techniques for emotional requirements. The extracted techniques were categorized and analyzed in terms of the elicitation approach, validation methodology, and domain of application. Table 9 depicts the elicitation techniques extracted from the literature reviewed. The following subsections discuss the categorization of the elicitation approaches, methods for validation, benefits, and limitations, and domain of application of these techniques.

 

Alkhomsan et al. (2024), *PeerJ Comput. Sci.*, DOI 10.7717/peerj-cs.1782

**Table 9  Categorization of emotional requirements eliciting techniques.**

| Category | Study | Used technique | Validation | Domain |
|---|---|---|---|---|
| Interviews and Workshops | Miller et al. (2015) | semi-structured interviews | Illustrative case study | Healthcare |
| | Thew & Sutcliffe (2018) | Interviews and Workshops | Illustrative case study | Information systems |
| Survey/Questionnaires | Xiang, Yang & Zhang (2016) | Survey of IS team members | Illustrative case study | Information systems |
| | Proynova et al. (2010) | Personal values elicitation | No Validation | Information systems |
| | Proynova et al. (2011) | Value and attitude questionnaires | Illustrative case study | Healthcare |
| | Cockton et al. (2009) | Sentence completion | Illustrative case study | Game Development |
| Scenarios/Stories | Maier & Berry (2017) | User stories classification | Experiment | Information systems |
| | Ramos, Berry & Carvalho (2005) | Storyboarding | Illustrative case study | Information systems |
| Prototyping | Callele, Neufeld & Schneider (2009b) | Iterative requirements capture process | No Validation | Game Development |
| Modeling/Mapping | Sherkat et al. (2018) | Emotional attachment framework | Experiment | Information systems |
| | Sutcliffe (2012) | User-oriented requirements process | Illustrative case study | Healthcare |
| | Bolchini, Garzotto & Paolini (2007) | Communication goals framework | Illustrative case study | Mobile applications |
| Machine Learning Techniques | Jean-Charles, Haas & Drennan (2019) | Supervised machine learning | No Validation | Information systems |
| | Jiang & Li (2020) | Multi-aspect sentiment analysis | Illustrative case study | Automobiles |
| | Cheng et al. (2023) | Multi-Modal Emotion Recognition | Experiment | Information systems |
| Analytical Techniques | Colomo-Palacios et al. (2010) | affective grid | Illustrative case study | Information systems |
| | Colomo-Palacios et al. (2011) | affective grid | Illustrative case study | Information systems |
| | Sutcliffe (2011) | Analysis of users' affective reaction | Illustrative case study | Information systems |
| | Dong, Guo & Liu (2014) | Fuzzy Cognitive Model | No Validation | Information systems |
| | Han et al. (2022) | Analyzing online reviews | Illustrative case study | e-commerce |
| | Scherr et al. (2019) | Biometric sensing via TrueDepth camera | Illustrative case study | Mobile applications |

Alkhomsan et al. (2024), *PeerJ Comput. Sci.*, DOI 10.7717/peerj-cs.1782

**Table 9** (*continued*)

| Category | Study | Used technique | Validation | Domain |
|---|---|---|---|---|
| | *Zhou, Jianxin Jiao & Linsey (2015)* | two-layer model for latent customer needs elicitation | Illustrative case study | Information systems |
| | *Mulvenna et al. (2017)* | Focus groups and storytelling | Illustrative case study | Well-being |
| | *Taveter et al. (2019)* | Interviews and workshops | Illustrative case study | Healthcare |
| | *Stade et al. (2019)* | Interviews, biometric sensing | Illustrative case study | Mobile applications |
| | *Jackson & Norta (2020)* | Remote elicitation feedback process | Illustrative case study | Healthcare |
| | *Hsu & Chen (2021)* | Questionnaires and focus groups | Illustrative case study | Self-Service Technology |
| | *Curumsing et al. (2019)* | Content analysis and affinity diagramming | Illustrative case study | Well-being |
| | *Taveter & Iqbal (2021)* | Motivational goal modeling | Illustrative case study | Healthcare |
| | *Abdullah et al. (2020)* | Work system design, user stories | Illustrative case study | Healthcare |
| | *Levy (2020)* | Customer journey mapping | Illustrative case study | Well-being |
| | *Callele, Neufeld & Schneider (2009a)* | Emotion markers | No Validation | Game Development |
| | *Callele, Neufeld & Schneider (2008c)* | Interviews, threat analysis | Illustrative case study | Game Development |
| | *Callele, Neufeld & Schneider (2006)* | Intent and context analysis | No Validation | Game Development |
| Hybrid Techniques | *Thew & Sutcliffe (2008)* | Taxonomy for soft issue analysis and Interviews | No Validation | Information systems |
| | *Callele, Neufeld & Schneider (2008a)* | Surveys, interviews | Illustrative case study | Game Development |
| | *Callele, Neufeld & Schneider (2008b)* | Surveys, interviews | Illustrative case study | Game Development |
| | *Yu et al. (2021)* | Conversational bots | Experiment | Information systems |
| | *Alkhomsan, Baslyman & Alshayeb (2022)* | Interviews, Prototyping, and Think aloud | Illustrative case study | Healthcare |
| | *Zulkifli & Shiang (2023)* | Motivational goal modeling | Illustrative case study | Learning |
| | *Lopez Lorca, Burrows & Sterling (2018)* | Motivational goal modeling | Illustrative case study | Learning |
| | *Iqbal et al. (2023b)* | do/be/feel and Motivational goal modeling | Case study and Experiment | Sociotechnical systems |

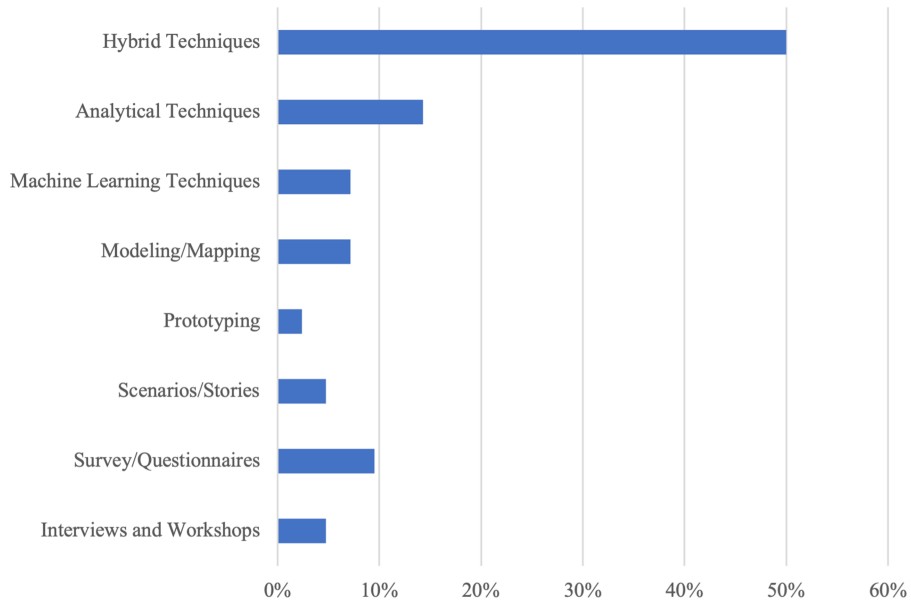

**Figure 4** **RE elicitation techniques categories.**

## Elicitation techniques categorization

### •Interviews and workshops

Interviews and workshops involve direct engagement with stakeholders through open discussions, focus groups, or other interactive sessions to elicit requirements. These techniques enable analysts to explore stakeholders' perspectives in depth through open-ended questions, brainstorming exercises, and collaborative activities. Key benefits of interviews and workshops include gaining contextual insights, uncovering hidden needs, and building bonds with stakeholders. However, results can be influenced by facilitator bias and limited participant availability.

Interviews and workshops were used in two studies (5% of the total), as seen in Fig. 4. *Miller et al. (2015)* conducted semi-structured interviews with emergency dispatch officers, asking them to describe challenging calls and the emotions experienced. Through qualitative analysis of these emergency call narratives, key emotions like stress, urgency, and empathy were identified. The interviews also used emotion-oriented questions to directly elicit affective needs and priorities from the participants. Thew and Sutcliffe carried out interviews and workshops and inquired about the values driving different user groups (*Thew & Sutcliffe, 2018*). By constructing value hierarchies and using laddering questioning techniques, they derived underlying emotional goals from the expressed values. These emotional goals were then mapped to quality attributes and more detailed requirements.

### • Survey and questionnaires

Surveys and questionnaires present stakeholders with standardized sets of questions to gather requirements data. Closed-ended questions and rating scales allow for quantitative analysis of stakeholder priorities, preferences, and concerns. Surveys provide an efficient

way to collect input from large, geographically dispersed groups. However, they limit the deeper discovery of stakeholder viewpoints found in interviews. Question biases also pose a threat to validity. Surveys and questionnaires accounted for four studies (10%). They were mainly validated *via* case studies, with applications in information systems, healthcare, and game development.

*Xiang, Yang & Zhang (2016)* surveyed team members for emotional requirements in an information system project. *Proynova et al. (2010)* and *Proynova et al. (2011)* proposed value and attitude-driven elicitation techniques based on mapping questionnaires to quality attributes. The conceptual approach is outlined (*Proynova et al., 2010*), while *Proynova et al. (2011)* represents an initial implementation and piloting. *Cockton et al. (2009)* applied sentence completion surveys to uncover player emotions, motivations, and values about gaming. The textual responses were analyzed to create worth maps of important qualities for players.

- **Prototyping**

Prototypes provide stakeholders with simulations of the final system to interact with and provide feedback. Low-fidelity prototypes focus on conceptual design and workflows, while high-fidelity prototypes offer near-complete functionality. Observing user interactions and emotions elicited by prototypes reveals experiential requirements. However, users may focus excessively on surface details. Prototypes also entail development costs and can raise unrealistic expectations of the final system. Prototyping was used in only one study (2%). *Callele, Neufeld & Schneider (2009b)* followed an iterative prototyping process to elicit emotional needs for games without validation, signaling a need for more research on this elicitation approach.

- **Scenarios/stories**

Scenarios and stories paint a narrative picture of how users will interact with the system. By contextualizing requirements in real-world situations, analysts can identify usability issues, edge cases, and emotional elements. However, scenarios rely on analyst perceptions and may miss unexpected use cases. Representing all stakeholder needs through scenarios can also prove challenging. This approach was utilized by two studies (5%). *Maier & Berry (2017)* adopted a user story elicitation approach, having participants provide open user stories for a messaging application. User stories were annotated manually to identify ones expressing hedonic/emotional quality *versus* pragmatic quality. *Ramos, Berry & Carvalho (2005)* used storyboarding to elicit stakeholder emotions, values, and beliefs during organizational transformation requirements engineering.

- **Modeling and mapping**

Models and maps provide abstract visual representations of stakeholders, systems, processes, or other elements. Flow charts, customer journey maps, goal models, and other diagrams help analysts elicit and organize requirements. Visual models also aid communication with stakeholders. However, complex real-world dynamics may not fully translate to models. Analysts must take care to keep models simple enough to understand and use.

Modeling and mapping techniques accounted for three studies (7%). *Sherkat et al. (2018)* proposed the Emotional Attachment Framework that adapted existing models of

emotional attachment to categorize drivers and map them to emotional goals in software design. This modeling enabled capturing and understanding emotional requirements. Sutcliffe used goal analysis to map obstacles to motivations and plan system responses to transform negative emotions into positive ones (*Sutcliffe, 2012*). This mapping of goals and emotions facilitated tracing the sources of affective requirements. *Bolchini, Garzotto & Paolini (2007)* proposed mapping brand values to communication goals and then deriving aligned requirements. This modeling linked brand concepts to requirements through intermediate communication goals.

- **Machine learning techniques**

Machine learning techniques apply algorithms to large datasets to uncover patterns and insights. For requirements elicitation, machine learning can analyze user behaviors, feedback, and communication to identify preferences. Benefits include finding trends in unstructured data that humans could miss. Challenges include needing sufficient training data, interpreting complex algorithms, and addressing biases in data.

Machine learning was rarely used for elicitation, with only two studies (7%) employing it in different domains. *Jean-Charles, Haas & Drennan (2019)* proposed using supervised machine learning techniques to convey the emotional state of stakeholders during requirements elicitation interviews in real-time. This would allow analysts to adjust questioning approaches for sensitive topics based on the predicted emotional range. However, specific machine learning methods and validation were not detailed. Jiang and Li developed a multi-aspect sentiment analysis method using machine learning for mining customer requirements from online reviews (*Jiang & Li, 2020*). This involved feature-sentiment word pair extraction, modifying an algorithm for enhanced accuracy, and converting sentiments into Kano categories. The technique was verified through real automobile review data, but limitations exist in handling reviewer value differences. Another recent study (*Cheng et al., 2023*) proposes a multi-modal emotion recognition platform that integrates facial expressions, vocal intonation, and textual sentiment analysis. It captures stakeholders' emotional cues in real-time.

- **Analytical techniques**

Analytical techniques apply mathematical and statistical methods to derive requirements insights from data. Key advantages are quantifying subjective qualities and uncertainties in requirements. However, solely relying on analytical techniques can overlook emotional and experiential aspects of requirements. Analytical techniques have been applied in (14%) of studies to uncover emotional requirements.

Affective grid analysis has been used to measure stakeholder emotions during requirements elicitation sessions, providing a quantitative analytical lens (*Colomo-Palacios et al., 2010*; *Colomo-Palacios et al., 2011*). Fuzzy cognitive models have analyzed relationships between customer emotions, perceptual patterns, and product design elements, enabling emotion-focused product optimization (*Dong, Guo & Liu, 2014*). Facial expression mapping to emotions *via* camera data has been proposed to enrich software requirements validation and verification, but further evaluation is needed (*Scherr et al., 2019*). Analyzing users' affective reactions to requirements and prototypes has been suggested to trace the source of emotions, improve specifications to remove negative

emotion triggers, reassure users, and transform negative emotions through system responses in applications like healthcare (*Sutcliffe, 2011*). Agent-based storyboards and scenarios illustrated the potential. Overall, analytical techniques demonstrate promise in eliciting emotional requirements but need expanded validation through additional case studies across problem domains.

- **Hybrid techniques**

Hybrid techniques combine two or more of the above elicitation methods to leverage their complementary strengths. For example, using surveys and interviews/workshops provides both quantitative insights and qualitative depth. However, hybrid techniques require expertise in multiple methods and can be resource-intensive to implement fully. Analysts must also carefully integrate divergent datasets from different techniques.

Hybrid techniques were most common in 21 studies (50%). They were validated mainly by case studies across diverse domains. This highlights the potential value of using mixed methods, but the need for expanded validation remains.

Structured interviews and workshops have been utilized to gather functional, quality, and emotional needs from users (*Taveter et al., 2019*). Interviews have also been supplemented with biometric sensing of emotions during system usage to connect user preferences and concerns to requirements (*Stade et al., 2019*). Customer journey mapping within a design thinking framework has helped uncover emotional needs through interviews and stakeholder feedback (*Levy, 2020*). Customized interviews, prototyping, and think-aloud techniques have identified emotional characteristics and goals for healthcare applications (*Alkhomsan, Baslyman & Alshayeb, 2022*). Content analysis of interviews, surveys, and trials combined with affinity diagrams and goal modeling has elicited smart home users' emotional goals (*Curumsing et al., 2019*). Conversational agents with multimodal sentiment analysis capabilities have been proposed to recognize user intentions and gather service requirements (*Yu et al., 2021*). Focus groups and storytelling have been used in a participatory design approach to elicit requirements from users with specialized needs, like people with dementia (*Mulvenna et al., 2017*). Surveys, focus groups, and statistical analysis have also revealed airport kiosk users' emotional needs (*Hsu & Chen, 2021*).

Across these studies, *Callele, Neufeld & Schneider (2009a)*; *Callele, Neufeld & Schneider (2008c)*; *Callele, Neufeld & Schneider (2006)*; *Callele, Neufeld & Schneider (2008a)*, and *Callele, Neufeld & Schneider (2008b)* developed a coherent line of research using interviews, surveys, and tools to elicit emotional requirements in video gaming contexts. First, *Callele, Neufeld & Schneider (2006)* developed emotional intensity maps to capture designer intent for player experiences. It proposed using focus groups with designers to elicit statements such as "I want the player to feel anxious as they approach the entry to this room." Another study introduced emotion markers to visualize requirements based on categorizing primary *vs* secondary emotions identified through interviews (*Callele, Neufeld & Schneider, 2009a*). Then, *Callele, Neufeld & Schneider (2008c)* is built on this by incorporating threat analysis and demonstrations to balance the potentially conflicting emotional and security needs of different stakeholders. Moreover, they used surveys and interviews to systematically analyze emotional and security requirements relationships in gaming (*Callele, Neufeld & Schneider, 2008a*). Finally, they enriched emotional requirements representation by

capturing contextual positioning, temporal and relational information in *Callele, Neufeld & Schneider (2008b)*. While proposing techniques for emotional requirements in games, these studies lacked empirical evaluation. Further research should evaluate the utility of these proposed approaches.

Emotional goal modeling has been fused with other techniques like remote elicitation and prototyping to capture emotionally driven requirements (*Jackson & Norta, 2020*). Combining motivational goal modeling with the theory of constructed emotion has been suggested to holistically elicit contextually grounded emotional requirements (*Taveter & Iqbal, 2021*; *Iqbal et al., 2023b*). In *Iqbal et al. (2023b)* the approach aims to represent stakeholders' emotional perspectives through iterative development and evaluation of prototypes, aided by appraisal meetings to build consensus on functional, quality and emotional goals. Techniques like work system analysis, user stories, and personas have also been combined with emotional goal modeling to elicit digital health system requirements (*Abdullah et al., 2020*).

Other proposed elicitation methods include sentiment analysis, association rule mining, and analogical reasoning to uncover latent customer needs from online reviews (*Zhou, Jianxin Jiao & Linsey, 2015*). Conceptual frameworks have also been developed to improve the elicitation and analysis of soft contextual factors like values, motivations, and emotions through interviews, surveys, and structured analysis (*Thew & Sutcliffe, 2008*).

## Methods for validating emotional requirements elicitation techniques

This section investigates the methods and approaches used to validate the proposed techniques of emotional requirements elicitation, as seen in Table 9. The results showed that 15% of studies proposed elicitation techniques but did not include validation. Most studies (76%) relied on illustrative case studies to demonstrate their approaches, as illustrated in Table 9. For example, in *Alkhomsan, Baslyman & Alshayeb (2022)*, the case study illustrated the proposed approach with 5 participants to elicit emotional requirements related to using virtual clinics within the healthcare app. The multi-method approach demonstrates the potential of emotion-oriented requirements engineering to uncover latent user requirements and enhance adoption by integrating emotional perspectives in goal models.

Only 9% of studies utilized experiments to empirically verify the proposed methods. For example, (*Yu et al., 2021*) conducted a controlled experiment using a conversational AI bot integrated with multimodal sentiment analysis to validate the effectiveness of incorporating sentiments for user intention mining.

## Benefits and limitations of emotional requirements eliciting techniques

Currently, the elicitation techniques have several limitations, which can be categorized into four general groups: time-consuming and human involvement, subjectivity, resource, and technique complexity, as seen in Table 10. Studies utilizing machine learning and sentiment analysis, such as in *Zhou, Jianxin Jiao & Linsey (2015)*, faced challenges with data sets, labeling, and the accuracy of human decisions in labeling specific reviews or sentences. Other studies proposed elicitation techniques requiring different resources, such

**Table 10  Limitations of emotional requirements eliciting techniques.**

| Theme | Study | Challenges |
|---|---|---|
| Time-consuming and human involvement | *Zhou, Jianxin Jiao & Linsey (2015)* | The supervised learning approach proposed in *Zhou, Jianxin Jiao & Linsey (2015)* requires substantial manual effort for labeling data to create training and testing sets. Additionally, the similarity-matching method to refine extracted product attributes still needs human judgment to determine attribute levels and hierarchies. |
| Subjectivity | *Ramos, Berry & Carvalho (2005)* | Identifying emotionally relevant requirements and achieving consensus can prove difficult for the requirements engineer when stakeholders have conflicting emotional perspectives, as subjectivity is inherent to emotion. |
| Resources | *Abdullah et al. (2020)* | The user stories approach in *Abdullah et al. (2020)* faced limitations in using language appropriately tailored and understandable for patients in the healthcare context. |
| | *Jiang & Li (2020)* | Expert knowledge in machine learning classification techniques is a prerequisite for applying the method proposed in *Jiang & Li (2020)*. |
| | *Yu et al. (2021)* | The machine learning technique proposed in *Yu et al. (2021)* relies on the availability of substantial datasets highly relevant to the specific application context and scenarios. |
| Technique complexity | *Cockton et al. (2009)* | Worth maps were difficult and confusing to some participants. |
| | *Maier & Berry (2017)* | User requirements are hard to classify as pragmatic or hedonistic. |

as experts in psychology. A psychological expert was needed for the technique proposed in *Sutcliffe (2012)*. In *Abdullah et al. (2020)*, clinic staff struggled to articulate emotional goals clearly. Knowledge of machine learning classification techniques was required for the technique proposed in *Jiang & Li (2020)*. Additionally, each review should be processed equally, and the technique's success depends on the quality of the reviewers.

The complex nature of emotions makes identifying requirements challenging like typical requirements. This explains the sophisticated methods used to elicit emotions. Several difficulties and confusion were associated with the techniques proposed in *Cockton et al. (2009)* and *Maier & Berry (2017)*. Some studies offered techniques more prone to subjectivity than others due to the lack of tools or systematic methods for handling emotions. For instance, the approach proposed in *Ramos, Berry & Carvalho (2005)* places a heavy burden on the requirements engineer to find relevant emotional requirements and satisfactory solutions, as some stakeholders have strongly differing emotions.

The benefits of the techniques discussed in the literature review include their ability to communicate emotions to stakeholders and consider diverse roles. A visual representation of emotional requirements was proposed as a straightforward way to convey emotional requirements to designers in *Dong, Guo & Liu (2014)*. The approach proposed in *Abdullah et al. (2020)* can capture emotional goals relating to different roles interacting with the system. Another approach considered improved user experience by recognizing user emotions and developing user-friendly conversational AI bots (*Yu et al., 2021*). To help

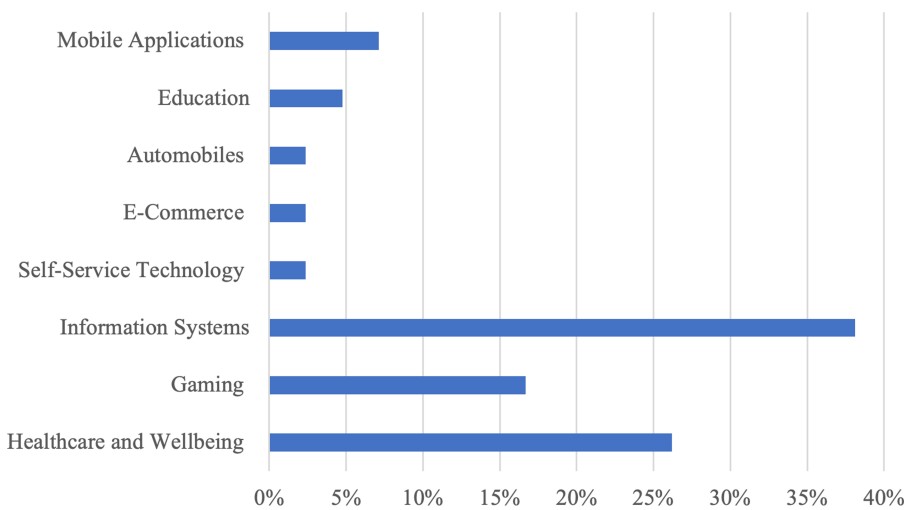

**Figure 5** Elicitation techniques domains of applications.

users express emotions and address challenges in describing emotions, an approach was proposed using a circumplex model built on valence and arousal dimensions (*Alkhomsan, Baslyman & Alshayeb, 2022*).

## Emotional requirements applications domains

The purpose of this section is to identify the domains in which emotional requirements are most commonly investigated. Several key issues will be discussed, including understanding the domains that tend to consider emotional requirements and determining which elicitation techniques may be reused or improved for similar domains. Figure 5 outlines the distribution of primary studies across various domains pertinent to emotional requirements elicitation. The healthcare and well-being domain encompasses 11 studies (26% of the total), highlighting the prominence of emotional requirements in systems intended for healthcare and well-being purposes. Gaming represents the second most prevalent domain, with seven studies (17%). 14 studies (38%) fall under the broad domain of general emotional requirements not confined to a particular application area. The remaining domains of self-service technology, e-commerce, education, automobiles, and mobile applications contain 1–3 primary studies each, representing 3–7% of the total studies.

## Emotional requirements modeling languages

Modeling is a crucial activity in requirements engineering to visually represent system goals, behaviors, and stakeholder needs. However, only a few studies have investigated modeling approaches tailored to emotional requirements, given their subjective and complex nature. This section discusses the requirements of modeling languages used to capture and convey emotional goals. Table 11. summarizes several studies that have proposed extensions or novel applications of modeling languages for capturing and analyzing requirements, with a focus on emotional and soft goals. The studies cover a diverse set of domains, including information systems, healthcare, learning, and video game development.

**Table 11   Requirements modeling languages used to model emotional goals.**

| | RML | Study ID | Domain of application |
|---|---|---|---|
| | SysML modeling extensions | *Schindel (2006)* | Information System |
| Extending existing modeling languages | Motivational goal modeling | *Miller et al. (2015)*; *Curumsing et al. (2015)*; *Taveter et al. (2019)*; *Jackson & Norta (2020)*; *Curumsing et al. (2019)* | Healthcare and Wellbeing |
| | | *Yeewai et al. ()*; *Zulkifli & Shiang (2023)*; *Lopez Lorca, Burrows & Sterling (2018)* | Learning |
| | | *Iqbal et al. (2023b)* | Sociotechnical system |
| | Goal-oriented Requirements Language (GRL) | *Alkhomsan, Baslyman & Alshayeb (2022)* | Healthcare and Wellbeing |
| Domain-Specific Modeling Languages | Videogame Emotion Language (VEL) | *Miguéis, Araujo & Moreira (2019)* | Game Development |

Motivational goal modeling has been used in several studies to represent stakeholders and their goals. The motivational power of emotions is leveraged in these models to drive requirement elicitation and design processes. In *Miller et al. (2015)*, the authors extended Sterling and Taveter's notion—motivational goal modeling notation—to capture emotions (*Sterling & Taveter, 2009*). The key extension proposed by the authors was the addition of "emotional goals" to capture the reflective-level emotions and desires of stakeholders. They defined two types - personal emotional goals for inherent user emotions and context-specific goals for system-evoked emotions. The notation for emotional goals involves connecting a role to an emotion and then to a functional or quality goal. This captures that the role has the associated emotion with respect to that goal. The People-Oriented Software Engineering (POSE) framework exemplifies this approach, adding simple heart symbols to agent-modeling diagrams to signify emotional elements indicating how stakeholders should feel when engaging with the system. The study distinguished between two types of emotional goals: personal emotional goals, which model the emotional desires of users, and context-specific emotional goals, which model the desired effects that a system has on its users. The strength of POSE models lies in their straightforward notation and emphasis on high-level concepts, which make them particularly useful in the early phases of requirements engineering. Moreover, POSE models have a greater value as shared artifacts in meetings to communicate with different stakeholders. Built on this work, the authors' group combined emotional goal modeling with viewpoint modeling, where they can capture the viewpoints of multiple stakeholders, including their emotional needs (*Curumsing et al., 2015*), which is extended by more recent work in *Goschnick (2018)*; *Sterling, Lopez-Lorca & Kissoon Curumsing (2021)*. They found that early-phase POSE models are most effective for understanding needs and applied this in a smart home system design in *Curumsing et al. (2019)*. In *Taveter et al. (2019)* and *Jackson & Norta (2020)* the POSE framework was also adopted for eliciting holistic emotional requirements. In *Taveter et al. (2019)*, the study uses POSE to elicit emotional requirements for two e-healthcare systems through interviews to capture emotions related to using the system. In *Jackson & Norta (2020)*, the study applies POSE remotely *via* an online questionnaire. The participants were asked about emotions in different usage scenarios. POSE Framework provides a means to conceptualize emotional

requirements; however, it does not provide an emotional requirements-specific analysis method.

Another study in the domain of interactive quiz applications presents an agent-oriented modeling approach (*Yeewai et al., 0000*). The approach comprises two phases—requirements elicitation and system design. The requirements phase focuses on understanding user needs through agent-oriented competency questions. The system design phase covers modeling interactive elements. Based on elicited responses, multiple-agent models are constructed to represent emotional factors like motivations.

Goal-oriented Requirements Languages (GRL) GRL was used to model emotional goals in relation to other goals that exist in the goal models (*Alkhomsan, Baslyman & Alshayeb, 2022*). The study first defined emotional requirements elements and then mapped them to the elements of the GRL model. Also, it proposed modeling emotional goals in a specific entity called an actor's emotions, which includes only an actor's emotions. As a result, requirements will be better managed, and concerns will be separated. Emotional goals are defined as soft goals linked to other goals or tasks through contribution links. Overall, these studies demonstrate the prevalence of extending proven goal modeling notations with emotion-oriented constructs, given their ability to formalize subjective concepts.

The study of *Schindel (2006)* suggests extending SysML to include a "Human View" for soft requirements elicitation. The approach proposes modeling logical subsystems representing human behaviors, including emotional aspects, using features, attributes, and couplings. These logical human models can capture emotional behaviors, outcomes, and requirements while avoiding the need to model their physical basis. The human model is then integrated with the technical product model within a unified SysML model. This allows emotional attributes to be tied to product features and behaviors affecting human experience.

Domain-specific modeling languages allow tuning modeling semantics and notation precisely for expressing emotions within a particular application context. A study proposed the Videogame Emotion Language (VEL) specifically for representing emotional requirements in games, implementing a prototype editor for constructing VEL models (*Miguéis, Araujo & Moreira, 2019*). This provides customized language constructs like "Emotions", "Objects", and "Triggers" tuned for modeling gaming experiences from an affective perspective. VEL was developed through domain analysis to identify core gaming emotion concepts, metamodeling to define abstract syntax, capturing key semantics and rules, and visually representing concrete syntax. It aims to help game developers understand and analyze player emotions and conflicts during initial design phases.

Overall, the prevalence of agent-oriented and goal-oriented modeling aligns with the focus on understanding stakeholder intentions, motivations, and emotions. However, model analysis, such as trade-off or conflict analysis, is not addressed.

### Emotional requirements modeling languages benefits and limitations

A variety of modeling languages have been proposed to represent emotional requirements, offering potential benefits as well as facing limitations. Extensions to SysML integrate requirements focused on human-experienced qualities, emotions, and subjective

perspectives (referred to here as "soft requirements") into systems models, enabling the capture of diverse expert perspectives in *Schindel (2006)*, but require further research for real-world applications. Approaches extending motivational goal modeling (*Miller et al., 2015*; *Curumsing et al., 2015*) allow simple graphical communication of emotional goals with stakeholders early in requirements engineering but risk introducing complexity through implicit relations. Alternative motivational goal modeling techniques link requirements and system design through competency questions; however, the multi-model process can be challenging to implement fully to derive designs (*Yeewai et al., 0000*).

Motivational goal modeling represents functional, quality, and emotional goals, roles, and relations to support communication throughout the design process, yet needs integration into development activities (*Taveter et al., 2019*). Domain-specific languages like VEL provide customized elicitation concepts and visual notation tailored to video games, though they involve extensive modeling overhead (*Miguéis, Araujo & Moreira, 2019*). Enhanced agent-oriented models can identify user concerns early and improve requirements completeness but require extension across problem domains (*Curumsing et al., 2019*). In summary, the key benefits of modeling languages are facilitating communication, while limitations include complexity, implementation difficulties, and insufficient validation across application contexts.

### Emotional requirements modeling languages applications domains

The healthcare and wellbeing domain leads with five studies employing agent-oriented and goal-oriented modeling to capture emotional requirements in applications like mental health, caregiving, and emergency response systems. Capturing emotional requirements appears highly valuable in these contexts where user experience and emotional outcomes are central. Beyond healthcare, RML-based emotional modeling also holds promise in information systems design, as in *Schindel (2006)* proposed extensions for incorporating "soft" user requirements into model-based engineering. Emotional factors may enhance user acceptance and adoption of enterprise systems.

Additionally, the gaming domain offers an expected application, given the significant role of experience, narrative, and emotion in games. This is evidenced by domain-specific Videogame Emotion Language proposed in *Miguéis, Araujo & Moreira (2019)*. Tailored vocabularies and semantics can capture the nuances of emotion in gaming scenarios.

## DISCUSSION

In this section, we discuss the results of this study under the elicitation technique and modeling language sections. In addition, we present some of the challenges associated with Emotional requirements elicitation and modeling. Finally, the section addresses some of the threats to validity and identifies interesting future research directions considering gaps and opportunities highlighted through this comprehensive literature review.

### Emotions in RE

The topic of ER is relatively new, with 46 studies published from 2006 to 2023. The majority of studies published between 2019–2023 aimed to increase users' acceptance and

trust in smart home systems for the elderly (*Curumsing et al., 2019*). Such solutions became a necessity to overcome issues caused by the aging population and the huge burden of hospitalization on the healthcare infrastructure. In other studies, machine learning (ML) techniques, specifically natural language processing (NLP), were widely employed to extract requirements from online reviews and facilitate the collection of requirements. Recently, ML techniques have been explored and utilized more for requirements elicitation since COVID-19 imposes a social distance, which hinders the work of requirements engineers (*Stade et al., 2019*). The findings indicate that healthcare and well-being contexts are the most prevalent because their design focuses on how they can meet the personal and social needs of users. Incorporating emotional requirements can ensure these systems effectively support human-technology interaction. Overall, emotional requirements have growing relevance across domains, especially healthcare, to address emerging needs. Continued adoption of ML techniques may further enable remote elicitation. Healthcare and well-being systems represent a prime area for ER research, given their focus on users' social and personal needs.

## Emotions definitions

The examination of these studies highlights some key issues and limitations in how emotions have been conceptualized and defined within requirements engineering research. It was noticed that many of the papers lack clear, rigorous definitions regarding the central construct of emotion. Most studies did not provide explicit definitions, and those that did offer some characterization that describe emotions in broad, imprecise terms such as subjective experiences or internal states.

The reviewed studies reveal several key conceptual perspectives on emotions in requirements engineering and human–computer interaction research. The valence dimension provides a useful starting point for differentiating between positive and negative emotions to understand their potential impact on users' experiences and reactions. However, this binary categorization has limitations in capturing the complexity of emotional states.

Incorporating arousal as an additional dimension aids in characterizing emotions based on intensity as well as valence (*Alkhomsan, Baslyman & Alshayeb, 2022*). However, some critics argue that discrete categories cannot fully represent the fluid, dynamic nature of emotions. Appraisal theories address this by defining emotions as context-dependent intuitive reactions shaped by subjective evaluations of events, agents, or objects (*Roseman & Smith, 2001*; *Sherkat et al., 2018*; *Callele, Neufeld & Schneider, 2009a*; *Callele, Neufeld & Schneider, 2008c*; *Callele, Neufeld & Schneider, 2006*; *Callele, Neufeld & Schneider, 2008a*; *Callele, Neufeld & Schneider, 2009b*; *Callele, Neufeld & Schneider, 2008b*). This aligns with psychological constructionist views.

The theory of constructed emotion further proposes that emotions emerge from the neurocognitive processing of goals, prior experiences, and situational factors (*Taveter et al., 2019*; *Taveter & Iqbal, 2021*; *Taveter & Iqbal, 2021*). Rather than fixed internal states, emotions are constructed transiently based on the interaction between the individual and their environment. This dynamical systems perspective resonates with trends in psychology

emphasizing contextualized, embodied cognition. Linking emotions to Norman's reflective level of design also highlights their close ties to sociocultural meanings, values, and purposes (*Miller et al., 2015*). The paper argues that explicitly incorporating this level in requirements engineering can better capture emotional goals. Differentiating personal *versus* context-specific emotions further underscores the need for a multi-perspective approach.

The reviewed studies show an evolution in how emotions are conceptualized, moving away from static, decontextualized models towards more dynamic, constructivist perspectives aligned with modern affective science. This reflects a shift from viewing emotions as fixed internal states to seeing them as complex experiences constructed through processing goals, contexts, and prior interactions. Synthesizing diverse conceptual lenses provides a more detailed and multi-layered foundation for engaging with emotions in technology design and requirements engineering. Further interdisciplinary integration can enhance approaches to capturing emotional factors that shape human-technology relationships. Overall, the field displays progress in developing more holistic, human-centered understandings of the role of emotions.

## Emotional requirements elicitation techniques

There was a wide variety of elicitation techniques for emotional requirements, with hybrid approaches being the most common (21 studies, 50% of the total). This suggests combining methods may help capture emotional requirements from multiple angles, rather than assuming any single technique is fully comprehensive. For example, the mix of direct techniques like interviews with indirect ones like biometric sensing provides flexibility to elicit emotions that may be difficult for users to express directly (*Stade et al., 2019*). In addition, Interviews, along with prototyping, help reveal emotions experienced during realistic user interactions rather than just self-reported emotions (*Alkhomsan, Baslyman & Alshayeb, 2022*). Interviews and surveys were most common but require demonstrating reliability, rather than assuming inherent efficacy. Given the complexity of emotions, comprehensive validation of any elicitation technique is infeasible. However, researchers should seek evidence for reliability and appropriateness through replicating studies across contexts. Machine learning techniques were rarely used with a count of three studies. ML has strong potential for emotion analysis but has yet to be widely explored for requirements elicitation. ML techniques can detect emotional cues and sentiments from textual data like requirements documents. Also, it can analyze large volumes of requirements data efficiently for emotions. More research is needed to demonstrate its capabilities in this area. For example, apply natural language processing to better detect emotions in text and study which ML methods are most effective for emotion extraction from requirements.

Current research reveals promising opportunities in the well-being and game development domains. In particular, the gaming domain is primed for advancing emotional requirements practices, as games intrinsically motivate strong user emotions. Testing and validating contextual elicitation techniques like prototypes and playtesting in this domain could strengthen best practices more widely. Additionally, the emerging domain of wellness technologies requires eliciting sensitive, emotional factors to drive adoption and effectiveness. The application area substantively impacts progress, with gaming's

affective emphasis necessitating urgent elicitation practice improvements *versus* industrial domains with strict usability requirements that may see slower adoption. Thus, rigorous validation efforts on gaming and wellness can demonstrate tangible benefits and develop evidence-based guidelines.

While case studies can provide initial practical demonstrations, more controlled experiments are needed to thoroughly examine and compare the effectiveness of techniques, as case studies alone cannot isolate impact. While case studies demonstrate real-world feasibility, experiments better assess causality between methods and requirements quality. Prioritizing empirical studies with both experiments and case studies can strengthen the utility evidence base for emotional requirements elicitation.

## Emotional requirements modeling languages

The modeling of emotional requirements provides several key benefits. First, it allows for analyzing how various emotional needs relate to and impact other functional and non-functional requirements. Mapping out emotions as explicit requirements enables designers to reason about how inducing certain emotions could enable or inhibit other goals and behaviors. For example, designing to promote feelings of trust may facilitate greater disclosure of personal information, while anxiety may inhibit usage altogether. Also, emotional requirements modeling enables traceability from these affective considerations to downstream design decisions. This supports the assessment of whether certain features are properly aligned with intended emotional outcomes. Design choices can be evaluated based on their consistency with specified emotional needs.

The modeling of emotional requirements presents inherent challenges due to the subjective, intangible nature of human emotions. Initial efforts have been made to adapt standard modeling techniques or develop customized languages specifically for representing emotional user needs and goals. A predominant approach is extending established goal-oriented and agent-modeling methods to incorporate graphical notations denoting emotional goals, as exemplified in the People-Oriented Software Engineering (POSE) framework proposed across multiple studies. This reflects a conceptual orientation toward depicting emotions from a stakeholder-centric perspective. The POSE framework's differentiation between personal and contextual emotional goals also provides a useful abstraction for modeling the various sources and targets of affective requirements. While POSE provides rich details on emotions, it lacks analytical capabilities. Goal-Oriented Requirements Language (GRL) has been extended to capture emotional goals and their relationships with other goals (*Alkhomsan, Baslyman & Alshayeb, 2022*). GRL can be further extended to conduct a quantitative analysis of how emotional goals contribute to other goals. GRL also can simulate how different design choices could impact satisfaction levels of emotional goals. A promising direction is to extend modeling languages by leveraging the complementary strengths of both qualitative emotion representations from frameworks like POSE and the quantitative goal analysis techniques provided by languages such as GRL. This would allow comprehensive modeling of emotions along with systematic analysis of their interdependencies and influences.

For highly interactive systems like games, domain-specific languages like the Videogame Emotion Language (VEL) allow tuning modeling vocabulary and semantics precisely to the application context to capture experiential requirements. However, rigorous empirical evaluation of the effectiveness of domain-specific emotion modeling languages remains limited. In contrast, viewpoint-oriented modeling shows promise as a general technique for eliciting comprehensive emotional requirements by explicitly capturing the perspectives of diverse stakeholders. Mapping these emotional goals onto standard goal modeling languages promotes interoperability with existing tools and processes, such as in *Alkhomsan, Baslyman & Alshayeb (2022)*.

## Challenges associated with developing er elicitation and modeling techniques

The complex nature of emotions makes it difficult for software engineers to build systems that reflect users' emotional goals. The key challenge is identifying and understanding people's subjective emotional needs, which are distinct from functional requirements and relate to desired feelings from using the software. Often, people struggle to articulate emotions directly. Although people can recognize their own emotions when experienced, they may find it hard to express them in advance since emotions are a subjective component of consciousness. Further, even if people communicate emotions, there remains uncertainty in how those emotions will be interpreted and transformed into specific requirements due to the ambiguous, unstructured, and difficult-to-quantify nature of emotional requirements.

Many techniques for eliciting emotions have limitations stemming from their inherent subjectivity or their complexity, time intensity, and need for human involvement. Requirements engineers can find themselves overwhelmed by the complexity of techniques, while project teams become overburdened by time-consuming tasks and limited resources. Per the study findings, elicitation techniques adapted from existing methods can be more effective regarding time, cost, and resources. However, without supporting tools, these techniques risk misinterpretation of the elicited emotions by the requirements engineer. Additionally, validating the effectiveness of elicitation techniques presents challenges. Most have only been demonstrated through illustrative case studies without validating their effectiveness and efficiency. This stems from the lack of metrics and the subjective nature of emotional requirements. Clear definitions of emotional requirements elements are needed first to validate a technique's ability to elicit those elements. As a result, most proposed techniques have only been assessed for potential and feasibility, which is insufficient to determine their efficacy and efficiency in eliciting emotional requirements.

Modeling emotions in requirements engineering remains a significant research challenge due to the inherently complex, subjective nature of emotional states and experiences. Two primary issues are representing the dynamic changes in emotions over time and contexts and quantifying emotional data for enhanced analysis and decision-making. A core modeling challenge is the lack of representations of how emotions evolve over time and in response to contextual triggers. More research is needed on capturing emotion dynamics. Factors like demographics, environment, and social influences can alter emotions, so models should incorporate these factors. Longitudinal studies are required to understand

emotion patterns, stability, and transitions. Also, capturing emotions through quantitative metrics poses significant research challenges for requirements modeling languages. It has important analytical benefits for requirements prioritization, impact estimation, and system optimization.

## Comparing results to related literature

To situate the findings of this review in the broader literature, it is informative to systematically compare the conclusions reached in our work to those of highly relevant prior reviews on emotional requirements engineering. In particular, the recent systematic mapping study by *Pacheco, García & Reyes (2018)* provides an opportunity for an in-depth contrast of the conclusions and future directions identified regarding elicitation techniques, modeling languages, challenges, and research gaps. *Iqbal et al. (2023a)* conducted a systematic mapping focusing specifically on categorizing and analyzing the state-of-the-art research on emotions in requirements engineering, providing an overview of the field. A subset of the reviewed studies targeted elicitation and modeling, which are the main focus areas in our review.

Both reviews identified a similar number of primary studies on emotional requirements engineering, with our review including 46 papers and Iqbal et al.'s review containing 34 papers. There was an overlap of 23 studies included as primary sources in both reviews, indicating substantial agreement on the core relevant literature. This demonstrates clear consensus on foundational literature, while diversity remains in peripheral sources depending on the review's precise scope, selected databases, and inclusion criteria.

For elicitation techniques, the categorization revealed strong parallels between the two reviews, with interviews, surveys/questionnaires, and hybrid approaches being dominant. Our review offers an enhanced analysis of limitations by categorizing them into four groups: time/human involvement, subjectivity, resources, and complexity. This structured understanding exceeded Iqbal et al.'s high-level discussion. Also, our review provides a clearer view of validation methods by quantifying the use of case studies (76%) and experiments (9%) to validate techniques. Iqbal et al. did not examine validation approaches in this level of detail.

For emotional requirements modeling, the two reviews exhibit close alignment. Extensions of goal-oriented and agent-oriented notations were central in both, enabling graphical representations of emotional goals and stakeholder perspectives. Domain-specific languages like the Videogame Emotion Language also emerged in each. The modeling technique similarities signify convergence on core approaches applied for emotional requirements to date. However, our review provides a more comprehensive coverage of modeling techniques with a dedicated focus on five studies, specifically on modeling languages. *Iqbal et al. (2023a)* had a broader scope without an in-depth focus on modeling.

Regarding application domains, healthcare, and well-being were dominant contexts in both reviews, underscoring the value of emotional requirements in these areas. Gaming was also a prominent domain in each. Some minor differences arose—our review included more enterprise/information system papers, while *Iqbal et al. (2023a)* covered more mobile

applications. But both agreed on health and gaming as the central domains that currently consider emotional requirements.

Finally, the overarching conclusions and future directions show strong parallels between the two reviews. Our identification of opportunities like elicitation tools and modeling dynamics offers more precise future directions compared to the high-level discussion by *Iqbal et al. (2023a)*. Further synthesis across reviews will continue to strengthen collective knowledge.

### Future directions

Users' emotions are a promising aspect that can augment the user experience if identified effectively. However, requirement elicitation and modeling that incorporate users' emotional needs remain limited. More effort is warranted to facilitate the integration of emotional requirements into practice. Considering our findings, we propose future research directions for eliciting and modeling emotional requirements.

Firstly, while some initial attempts have been made, eliciting emotional requirements remains challenging due to complexity, subjectivity, and resource constraints. Opportunities exist to investigate novel elicitation techniques and holistic ways of capturing and conveying ER. Validating collected ER also poses difficulties, as individuals perceive experiences uniquely. Resolving conflicting user emotions, especially across cultures, merits exploration. Developing tools to assist in eliciting emotions could prove beneficial, given the intricacy of feelings and the challenges of finding descriptive words universally. Such tools may facilitate cross-cultural, language-independent emotional expression. Secondly, modeling and systematically analyzing ER in relation to other requirements warrants examination. Bridging the gap between eliciting emotional goals and implementing design solutions requires a methodical approach to identify options that account for user emotions. Further research might evaluate ER-driven systems to quantify the value of considering ER during development.

Finally, additional domains that could benefit from emotional considerations include government, education, and e-commerce. Determining which emotions users wish to feel or avoid could enhance the system delivery of core value propositions across industries. Exploring privacy concerns and acceptance of emotion collection mechanisms also offer critical future directions. In summary, our findings highlight promising opportunities to advance the integration of users' emotional needs through collecting and modeling software requirements. Furthermore, to explore the full potential of emotions to augment systems and enhance user experiences across domains.

## THREATS TO VALIDITY

In this section, we discuss how threats undermine the validity of research results but were addressed through rigorous procedures. Several types of validity were considered, including internal, external, and construct validity. Threats to each type were identified, and steps taken to limit impact are discussed in the following.

### Internal validity

Internal validity refers to the integrity of the research methodology (*Kitchenham & Brereton, 2013*) and analysis in representing the topic evaluated. This includes the search strategy, which significantly impacts the results. A rigorous approach developed an accurate string using keywords published in papers. However, the search string may have limitations in capturing all relevant papers, indicating compromised internal validity. This was mitigated by snowballing references to find additional papers and discussing the impact on results. Researcher bias threatened study selection and data extraction. Two inclusion/exclusion phases (title/abstract, full-text) and a data extraction sheet mitigated bias.

### External validity

External validity considers how well findings can be generalized. The search was limited to four databases, potentially missing relevant studies published elsewhere. The threat was addressed by searching references for additional resources (snowballing). Furthermore, the exclusion of book chapters and technical reports from the search may have limited discovery of relevant practices and reduced generalizability.

### Construct validity

Construct validity examines if the review methodology measures what it intends to. In some studies, information to assess quality, methodology, or tool/method details was implicit, requiring subjective interpretation and reducing construct validity. Two researchers conducted quality assessments and resolved differences through discussion to limit subjectivity.

## CONCLUSION

Emotional requirements have gained more attention, and many studies have investigated how to incorporate emotional requirements into software engineering. As part of this systematic mapping, we have attempted to cover all the trends and areas in which emotional requirements have been utilized. During the study, elicitation techniques, modeling techniques, and application domains of emotional requirements were examined. In addition, the elicitation techniques were evaluated in terms of their benefits and limitations. According to our results, emotional requirements are a newly developed area that requires a lot of in-depth research into how to identify users' emotions. A total of 41 out of 46 selected studies either proposed or customized elicitation techniques to collect emotional requirements. A total of 12 studies have examined the modeling of emotional requirements, but many considerations remain to be addressed. In addition, the results emphasized the limitations of the elicitation techniques, which were categorized into four broad categories: time-consuming and human involvement, subjectivity, resources, and technique complexity. Finally, a set of open issues and research directions were presented as a result of this review.

## ACKNOWLEDGEMENTS

We acknowledge the comments and suggestions made by the anonymous reviewers to improve the manuscript.

### Funding

This article is published under funded project # INFE2204 from the Interdisciplinary Research Center for Finance and Digital Economy, King Fahd University of Petroleum and Minerals, Dhahran, Saudi Arabia. The funders had no role in study design, data collection and analysis, decision to publish, or preparation of the manuscript.

### Grant Disclosures

The following grant information was disclosed by the authors:
Interdisciplinary Research Center for Finance and Digital Economy:  INFE2204.
King Fahd University of Petroleum and Minerals, Dhahran, Saudi Arabia.

### Competing Interests

The authors declare there are no competing interests.

### Author Contributions

- Mashail N. Alkhomsan conceived and designed the experiments, performed the experiments, analyzed the data, prepared figures and/or tables, authored or reviewed drafts of the article, and approved the final draft.
- Malak Baslyman conceived and designed the experiments, performed the experiments, analyzed the data, prepared figures and/or tables, authored or reviewed drafts of the article, and approved the final draft.
- Mohammad Alshayeb conceived and designed the experiments, performed the experiments, analyzed the data, prepared figures and/or tables, authored or reviewed drafts of the article, and approved the final draft.

### Data Availability

 This is a literature review.

### Supplemental Information

Supplemental information for this article can be found online at http://dx.doi.org/10.7717/peerj-cs.1782#supplemental-information.

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
