# Peer review of "Eliciting and modeling emotional requirements: a systematic mapping review"

_PeerJ Computer Science, doi:10.7717/peerj-cs.1782_

## Round 0.1 · original submission · Major Revisions

Most important for your revision should be the restructuring and re-analysis of the results. Find better ways to group, discuss and contrast the existing work. Furthermore, please include a discussion on the new mapping study and how it compares to your results.
Please also take all the other reviewer comments into account for the revision. I believe they are valuable for improving your paper.
All in all, this will need considerable changes such that I will send the revision out for review again.

Reviewer 1 ·

Basic reporting

The intent of the review is spot on. There has been an increase in interest in emotional requirements in software engineering. So a well written review would be timely. I am not aware of one that is comprehensive and/or authoritative.

The survey is clear, well-written and follows an established methodology (more on that below).

Some definitions are missing, the most notable being what is an emotion. Defining emotions is a difficult task, one that would generate disagreement among researchers. Exactly what the authors mean by a framework and a process is also not clear, and is also expanded upon below.

Experimental design

The authors follow the approach of Kitchenham to attempt to generate an unbiased and comprehensive set of papers that they then analyse. They give a high level description of how papers were excluded, but it was insufficient to understand how it worked.

The categories by which they analyse the paper seem ad-hoc. The explanation for why they were chosen is not made clear.

Validity of the findings

The authors end up with 42 papers to analyse. Listing those 42 papers seems of some value. On the surface the papers constitute an interesting and representative sample of papers on emotional requirements.

However the analysis assumes the papers are independent. They are not.
Several are from the same research group., or are from researchers that were once part of a research group and now have evolved the research.
I can write with authority as I am author on several papers discussed and am aware of the lineage of others.

The count of papers on the topic or discussion of how the work splits between the topics is not significant.

Much better would be a comparison of the different approaches and an attempt to group related work, and point out contrasts between different groups. There is no such analysis or comparison.
The findings are of little value without that extra context.

Further the discussion about suggestions for future work discussion is not valuable without the context. There are different assumptions between the research groups which the authors do not take into account.

Additional comments

The authors have identified an important and interesting topic and attempted to address it systematically. More work needs to be done on the assumptions, about emotions, for example,
and the different approaches between different groups which refer to emotions differently.

Cite this review as

Reviewer 2 ·

Basic reporting

The review is broad but misses in Section 2 an important part of recent related work, which makes the review partially obsolete:

Iqbal, T., Anwar, H., Filzah, S., Gharib, M., Mooses, K., & Taveter, K. (2023). Emotions in Requirements Engineering: A Systematic Mapping Study. Proceedings of the 2023 International Workshop on Cooperative and Human Aspects of Software Engineering (CHASE 2023), 14-15 May 2023, Melbourne, Australia. IEEE (in press). Currently available as arXiv preprint arXiv:2305.16091.

The introduction does not clearly state if the SLR is created from the perspective of Human-Computer Interaction or Requirements Engineering. This confusion continues throughout the whole paper: the paper has also considered the works on the emotions and moods by software developers, which is different from the treatment of emotional requirements for software.

Experimental design

The search string that was used, which is represented in Table 1, seems to be too simple and does not directly include important terms like “requirements engineering”, “requirements elicitation”, “requirements analysis”, “requirements specification”, “requirements modelling”, etc.

The methodology should be more thoroughly described. There is no description of how duplicates were handled. In my opinion, having just binary quality scores 0 and 1 is not sufficient.

The division of the problem domain into the research areas organised into Sections 4.2.1-4.2.5 is not logical because it is too diverse, putting together “requirements engineering activities”, “user acceptance”, and “user experience” on one hand and “game development” on the other hand, which is not sensible. The research areas should be, for example, healthcare, information systems, mobile applications, smart home, game development, etc.

The categorization of the requirements elicitation techniques organized into Sections 4.3.1-4.3.5 is not logical because “customization of current elicitation techniques”, “machine learning”, “guidelines and frameworks”, “tools” and “methods” are too diverse and partially overlapping. Requirements elicitation techniques are generally known as, for example, interviews, questionnaires, workshops, observations, document analysis, literature reviews, obtaining sensor data, etc.

Validity of the findings

With respect to Section 4.4, emotional requirements elicitation techniques cannot be validated by emotional requirements modelling languages. These are two different categories which are not related to each other in the way claimed by the authors. The first category consists of instruments used to elicit emotional requirements and the second category consists of artefacts (including languages) used for representing emotional requirements.

The treatment of threats to validity in Section 5.6 is not thorough enough – it should explicitly address internal threats, external threats, and construct threats.

Additional comments

In Section 4.5, the authors should be careful when citing PS19 because it gives the impression that Role, Goal, Scenario, Interaction, and Behaviour models and Motivational scenarios were invented by this paper’s first author, while the truth is that these terms were coined in the reference 19, which is never referenced in PS19.

In my opinion, the Remote Emotional Requirement Elicitation Feedback Method (REREFM) gets too much attention in the paper considering that it is only a secondary method that is based on the POSE Method coined in the reference 18.

Cite this review as

---

## Round 0.2 · Major Revisions

The manuscript greatly improved by the revision. Yet, we identified a number of issues that still remain. Most importantly, the discussion of the paper by Iqbal et al. that is very close to your contribution needs to be more thoroughly discussed. Yet, there are several further concerns on the thoroughness of the analysis and the validity of the results. These need to be clearly addressed before publication.

Reviewer 1 ·

Basic reporting

The paper is considerably improved from the previous version. The authors have taken into account the feedback of both reviewers.

Experimental design

The paper list is not comprehensive in my opinion. However how the papers were chosen has been better explained. The collection of 42 papers is of sufficient interest.

Validity of the findings

The conclusions cited at the beginning of the paper are contestable.
In my opinion, there is a detailed development approach hfor emotional requirements, and the research is validating and improving it.
Also, an important future research area alluded to in the paper but not in the conclusions is managing conflicting emotional requirements. There is more critical than some other topics mentioned.

Additional comments

Small suggestions:

Change the title to Eliciting and Modelling Emotional Requirements ...
It reads better than Elicitation and Modelling Emotional Requirements ...

In section 2.1, Iqbal et al. [9] conducted the systematic mapping should be Iqbal et al. [9] conducted a systematic mapping

Reference 3 is incomplete – the paper is untraceable as is, and should be replaced or the complete citation found

Reference 27 has been superseded by Sterling, L., Lopez-Lorca, A. and Kissoon-Curumsing, M. (2018). Adding Emotions to Models in a Viewpoint Modelling Framework from Agent-Oriented Software Engineering: A Case Study with Emergency Alarms, in S. Goschnick (Ed.), Innovative Methods, User-Friendly Tools, Coding, and Design Approaches in People-Oriented Programming. Hershey, PA: Information Science Publishing, pp. 324-367
It was a rewrite of the paper for a book collection from chosen papers from Int. J. People-Oriented Programming.

Acknowledge that the anonymous reviewers improved the paper.

Cite this review as

Reviewer 2 ·

Basic reporting

The paper has improved but does not put its contributions in an appropriate context. Namely, on page 9, rows 101-102 the authors claim that "A systematic mapping of research on emotional requirements elicitation and modeling has not yet been conducted", which is not true, because a thorough systematic mapping study has been conducted by Iqbal, Anwar, et al. (2023) and has been published in the following publication (and NOT as an arXiv preprint paper as it is currently referenced in the manuscript): Iqbal, T., Anwar, H., Filzah, S., Gharib, M., Mooses, K., & Taveter, K. (2023). Emotions in Requirements Engineering: A Systematic Mapping Study. In 2023 IEEE/ACM 16th International Conference on Cooperative and Human Aspects of Software Engineering (CHASE) (pp. 111-120). IEEE. Please change the reference!

The language of the paper has improved but needs further polishing. For example, there is a double phrase "unknown-unknown" in the abstract, there is the "Finally, identify the..." phrase in row 239, omitting "we will", mistaken use of capital letter in row 679, etc.

Experimental design

The emotion definition groups are not adequate, and the papers are not correctly categorized according to the emotion definition groups. Emotions in requirements should rather be categorized according to the underlying groups of emotion theories, which are basic emotion theories, appraisal emotion theories, and the theory of constructed emotion. The currently used categories "emotions defined as subjective human experience" (or "feelings of users" in Table 8) and "emotions defined in relation to system qualities") overlap too much and are therefore not useful because emotions are always included in requirements with the purpose to increase the system quality. For example, according to Table 10 that contains studies that defined emotions in relation to system qualities, the source [35] treats emotions as "subjective experiental descriptions elicited from customers" and the source [43] as "subjective qualitative user reactions", which are not different from "emotions defined as subjective human experience". I recommend merging Tables 9 and 10.

In addition, there are some misunderstandings in Table 9. First, in the references [16] and [27], emotions are defined as “abstract human goals” rather than “explicit graphical modeling elements” (these are not definitive and therefore are of secondary importance). Later, the same definition of emotions as “abstract human goals” was also overtaken by the work referenced in [12]. Second, the references [19] and [23] are based on the theory of constructed emotion and NOT on psychological appraisal theory. Also, for the sake of completeness, the following reference should also be added to [19] and [23]: Iqbal, T., Marshall, J. G., Taveter, K., & Schmidt, A. (2023). Theory of constructed emotion meets RE: An industrial case study. Journal of Systems and Software, 197, 111544.

The two last research questions are vaguely stated. The first research question “What are the emotional requirements modeling languages and analysis that have been investigated in the studies?” should be rather formulated as ““What are the emotional requirements modeling languages and representation formats that have been investigated in the studies?” The last research question should rather be formulated as “What are the applications domains for which emotional requirements have been elicited and represented?”.
Based on my own experience, I do not think that conducting a pilot run on just five papers is sufficient to validate the study method.

Validity of the findings

Regarding the comprehensiveness and validity of the findings, I have the following remarks:
- In addition to the publication [12], emotional requirements for smart home systems for the elderly have also been addressed in the following journal paper, especially in the context of COVID-19 pandemic, which is mentioned in rows 617-618 of the paper. Therefore, the following publication should be included in the mapping study: Mooses, K., Camacho, M., Cavallo, F., Burnard, M. D., Dantas, C., D’Onofrio, G., ... & Taveter, K. (2022). Involving older adults during COVID-19 restrictions in developing an ecosystem supporting Active Aging: Overview of alternative elicitation methods and common requirements from five European Countries. Frontiers in Psychology, 13, 818706.
- The paper Mooses, et al. (2022) mentioned above should also be mentioned in rows 611-612, considering that the mapping study includes publications from the year 2022.
- The section “Interviews and Workshops” starting in row 304 should also explicitly include the “do/be/feel” workshop format of motivational modelling, which has been widely used for the elicitation of emotional requirements. The relevant reference is the following one: Lorca A. L., Burrows R., and Sterling L. (2018). Teaching motivational models in agile requirements engineering. In IEEE 8th International Workshop on Requirements Engineering Education and Training (REET), 30-39. IEEE.
- To avoid confusion, please use throughout the paper, including in Table 13, “motivational goal modeling” rather than “agent-oriented modeling”.
- In row 441, please correct the sentence so that it would begin as “Combining motivational goal modeling with the theory of constructed emotion…” and add the following reference, in addition to the reference [23], for the sake of completeness: Iqbal, T., Marshall, J. G., Taveter, K., & Schmidt, A. (2023). Theory of constructed emotion meets RE: An industrial case study. Journal of Systems and Software, 197, 111544.
- In Table 11, “No Validation” is not correct for the reference [23] because this paper uses for validation even two illustrative case studies.
- Please include the above reference Iqbal, Marshall, et al (2023) in Table 11 because it uses a real-life case study rather than just an illustrative case study.
- The term “soft requirements” used in row 575 has not been defined and therefore creates confusion.

The Discussion part of the paper should include a thorough and systematic comparison with the results obtained in the first mapping study of elicitation and modeling of emotional requirements by Iqbal, Anwar, et al. (2023) because the similarities and differences in the respective findings of the two quite similar literature reviews are of primary interest for the readers of your paper.

Additional comments

No additional comment.

Cite this review as

---

## Round 0.3 · Minor Revisions

While we see the paper as overall acceptable, we see the following issues necessary to be resolved in the revision:
- Revisit the criticism of existing elicitation methods to be more nuanced.
- Limitations that led to missing papers initially needs to be explicitely pointed out.

Reviewer 1 ·

Basic reporting

I have previously acknowledged this as satisfactory. The paper continues to improve.

Experimental design

The authors have included additional relevant papers that the reviewers have pointed out. That these exist points out a limitation of the study design. As the authors noted, they failed to reference the key paper describing the do/be/feel method which has been extensively used for elicitation of emotional requirements due to the lack of an appropriate keyword.

Also ignoring book chapters has limited the discovery of some broader applications. Perhaps this can be acknowledged again in the paper.

Validity of the findings

In my previous review, I described some of the claims as contestable. That remains the case.

For example, the authors have described the evolution of description of emotions.
As they write, "This reflects a shift from viewing emotions as fixed internal states to seeing them as complex experiences constructed through processing goals, contexts, and prior interactions."
Yet they criticise some methods of eliciting emotions, "For example, the studies that used interviews and surveys did not comprehensively validate if these techniques accurately capture complete, valid emotional requirements compared to alternatives. Their efficacy was assumed but not thoroughly demonstrated."
If emotions are complex experiences, it will not be possible to comprehensively validate if the methods accurately capture emotional requirements, and there is no sense that the emotional requirements are complete. How can it be possible to describe a complete set of what a person may experience? There is not exact agreement in how people use emotion words. I am sure my experience of happiness is different to that of the authors, as is my experience of fear, or sadness, or anger. So if the objective is that a paper elicits a happy response, what is correct? An image from a war may be happy for one side and sad for the other. Cat pictures are cute for some people and annoying for others.
So if I specify a requirement that a Web interface should elicit a feeling of cuteness, should cat pictures be included, and how will the requirement be validated.

So any critique of validation needs to be nuanced. The authors would be well advised to soften the critiques of methods.

Additional comments

Again, I commend the authors for taking the review comments seriously. The field of emotional requirements is still evolving. While I disagree with some of the perspectives of the authors, the paper can add to the discussion.

Cite this review as

---

## Round 0.4 · accepted · Accept

I have evaluated the new revision and found that the authors addressed all remaining comments from the previous round of reviews satisfactorily. I cannot see anything preventing the paper from being published now.